# Virtual screening of antimicrobial plant extracts by machine-learning classification of chemical compounds in semantic space

Hiroaki Yabuuchi[1¤]*, Kazuhito Hayashi[1], Akihiko Shigemoto[2], Makiko Fujiwara[1], Yuhei Nomura[2], Mayumi Nakashima[2], Takeshi Ogusu[1], Megumi Mori[1], Shin-ichi Tokumoto[2], Kazuyuki Miyai[1]

1 Department of Pharmaceutical Industry, Industrial Technology Center of Wakayama Prefecture, Wakayama, Japan, 2 Department of Digital Manufacturing, Industrial Technology Center of Wakayama Prefecture, Wakayama, Japan

¤ Current address: Kushimoto Branch, Shingu Health Center of Wakayama Prefecture, Wakayama, Japan
* yabuuchi_h0002@pref.wakayama.lg.jp

## Abstract

Plant extract is a mixture of diverse phytochemicals, and considered as an important resource for drug discovery. However, large-scale exploration of the bioactive extracts has been hindered by various obstacles until now. In this research, we have introduced and evaluated a new computational screening strategy that classifies bioactive compounds and plants in semantic space generated by word embedding algorithm. The classifier showed good performance in binary (presence/absence of bioactivity) classification for both compounds and plant genera. Furthermore, the strategy led to the discovery of antimicrobial activity of essential oils from *Lindera triloba* and *Cinnamomum sieboldii* against *Staphylococcus aureus*. The results of this study indicate that machine-learning classification in semantic space can be a highly efficient approach for exploring bioactive plant extracts.

## Introduction

Plant extracts have been used to treat various diseases for thousands of years. In eastern medicines, plant extracts have formed the basis for traditional medicine systems. In western medicines, by contrast, the isolation of bioactive low-molecular-weight compounds such as morphine (from opium), quinine (from cinchona tree), atropine (from *Atropa belladonna*) led to the idea of chemical compounds as drugs [1]. Identification of the active ingredients accelerated pharmacological researches, resulted in discovery of the target proteins and disentanglement of the molecular mechanism of actions.

Knowledge accumulation on active compounds has come with the development of information-rich approaches for efficient drug discovery. Quantitative structure-activity relationship (QSAR) and machine learning have been introduced to the drug development [2]. With the pharmacological reports increased, data resources for bioactive compounds such as MeSH, PubChem [3] and ChEBI [4] were also made available.

**Data Availability Statement:** R scripts and preprocessed literature data are available at https://github.com/yabuuchi-hiroaki/webvs. All other

relevant data are within the paper and its Supporting Information files.

**Funding:** This research was supported by the Kayamori Foundation of Informational Science Advancement (K32 ken XXV 577). The funders had no role in study design, data collection and analysis, decision to publish, or preparation of the manuscript.

**Competing interests:** The authors have declared that no competing interests exist.

Exploring novel medicinal plants is a major task in natural product research. In order to predict biological activity of the plant extracts, a mathematical model called quantitative composition-activity relationships (QCAR) was proposed [5, 6]. QCAR accounts the relationship of magnitude of the various chemical compositions of plant extracts with the bioactivity. However, its application to medicinal plant screening is limited because of (1) lack of the large-scale open data treating relation between composition and bioactivity of plant extracts, (2) difficulty in comprehensive compositional analysis covering diverse secondary metabolites in a plant sample, (3) necessity of composition data for all plant extracts to be predicted.

To circumvent these limitations, we have shown that a new computational screening strategy, word embedding-based virtual screening (WEBVS), has the potential to identify bioactive plant extracts. The overview of WEBVS is shown in Fig 1. Word embedding is known to encode semantic and syntactic similarity insofar as the embeddings for similar words will be nearby one another in vector space [7]. The WEBVS method utilizes the word embedding and a large amount of biomedical literature data to encode all known compounds and plants into a semantic space. The compounds are labeled by the presence/absence (active/inactive) of biological annotation data, and the labels and vectors are learned to construct a classification model. Finally, the labels of plants are predicted by the model in the semantic space. In this research, WEBVS was applied to screening of antimicrobial plant extracts, and was evaluated by statistical methods and antimicrobial assay against *Staphylococcus aureus*, a major human pathogen that causes a wide range of clinical infections [8].

## Materials and methods

### Data

Biomedical literature data with automatic annotation of chemical compounds and species was retrieved from Pubtator FTP site in September 2020 [9]. Biological annotation data of chemical

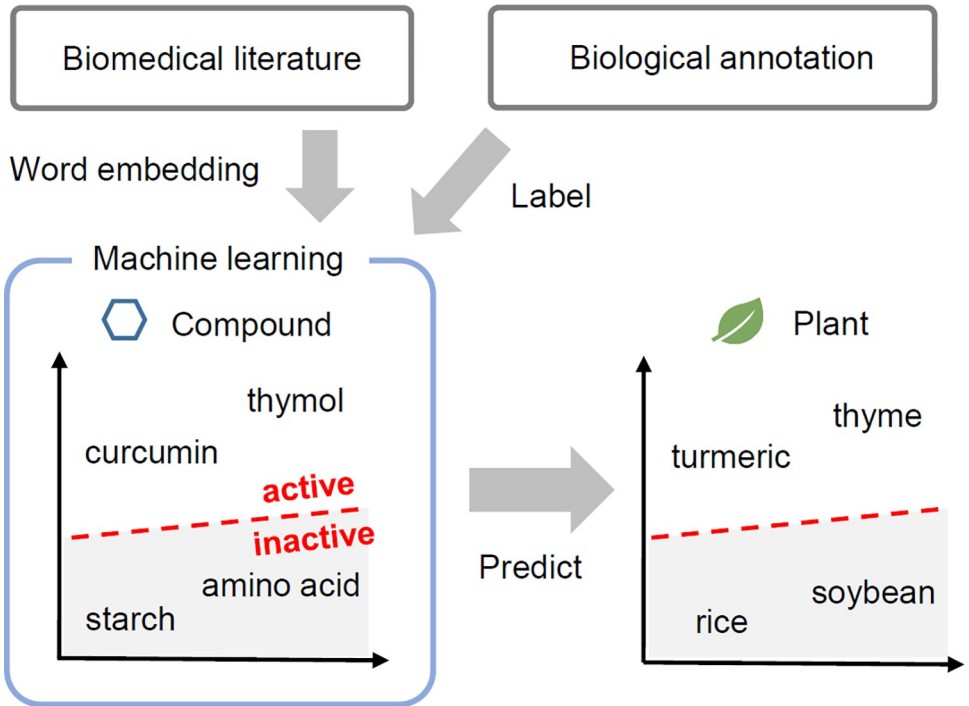

**Fig 1. Overview of word embedding-based virtual screening (WEBVS).**

compounds was retrieved from MeSH [3] and ChEBI [4] in September 2021. Plant taxonomic data was retrieved from NCBI Taxonomy [3] in September 2021. A list of antibacterial plants was retrieved from a systematic review conducted by Chassagne *et al.* [10] to evaluate prediction performance of WEBVS.

## Reagents

Acetone for gas chromatography was purchased from KISHIDA CHEMICAL Co., Ltd, Japan. Dimethyl sulfoxide (DMSO) and thymol (special grade) were purchased from FUJIFILM Wako Pure Chemical Corporation, Japan. A series of *n*-alkane standards ($C_9$ to $C_{40}$) was purchased from GL Sciences Inc., Tokyo, Japan. Mueller-Hinton II broth was purchased from Becton, Dickinson and Company, USA. *Staphylococcus aureus* (NBRC 12732) for antibacterial activity tests were from the National Institute of Technology and Evaluation, Biological Resource Center (NBRC), Japan.

## Preprocessing text data

We selected natural compounds annotated by "pharmacological action" term with "anti-bacterial agents" or "antifungal agents" or "fungicides, industrial" or "antitubercular agents" or "antibiotics, antitubercular" or "anti-infective agents" from MeSH, and those annotated by "has_role" relation with "antibacterial agent" or "antibacterial drug" or "antifungal agent" or "antifungal agrochemical" or "antifungal drug" or "antiinfective agent" or "antimicrobial agent" or "antiseptic drug" or "antitubercular agent" or "fungicide" from ChEBI. These compounds (128 compounds, S1 Table) were regarded as "active compounds" in this research. The other MeSH compounds were assumed to be "inactive compounds".

The biomedical literature data consisted of 132962 PubTator articles which contain both a bioactivity-related keyword ("activity", "action", "effect", "property", "efficacy" or "assessment") and a name of either active compounds or plants in their titles. The plant species, subspecies and variants were grouped at the genus level. Low-frequent words (appeared in less than 0.1% of the selected articles) and stop-words were removed from the abstracts of the articles.

## Word embedding

12356663 words appeared in the abstracts were inputted to word2vec embedding with continuous bag of words (CBOW) [11] to encode 16381 unique words as numerical vectors. "word2-vec" R package (version 0.3.4) was used for the embedding implementation. The number of dimensions was set to 100, the window size was set to 5, and the number of negative samples was set to 5.

## Machine learning of antimicrobial activity of chemical compounds

The embedded vectors of 128 active and 6443 inactive compounds were inputted to machine learning algorithms to classify the presence/absence of antimicrobial activity. As the labels of inactive compounds were uncertain, we randomly selected the same number of inactive compounds as that of active compounds. This selection was repeated ten times to avoid bias and increase robustness. Support vector machine (SVM) with the radial basis function kernel [12], random forest [13] and deep neural network [14] were tested by five-fold cross-validation with hyper-parameter optimization. The machine learning algorithm which showed the best accuracy was chosen as the best classifier. The labels of all embedded compounds were predicted

by the classifier, and were sorted by output probability of the presence of antimicrobial activity (hereinafter referred to as "antimicrobial probability").

## Virtual screening of antimicrobial plants

In order to predict labels of the plants, 2534 plant genera encoded by the word embedding were inputted to the classifier. The antimicrobial probability (classified as active if the value is above 0.5) was checked against the list of antibacterial plants, and plotted as an enrichment curve. "chemmodlab" R package (version 2.0.0) was used for plotting the curve with simultaneous plus-adjusted sup-t confidence bands [15]. Furthermore, two plants classified as active were selected for essential oil extraction, gas chromatography/mass spectrometry (GC/MS) analysis and antimicrobial assay.

## Extraction of essential oils

Fresh plant samples of *Lindera triloba* (syn. *Parabenzoin trilobum*) were collected from Koya town (Wakayama, Japan) in September 2021, and were separated into leaves and branches. Fresh plant samples of *Cinnamomum sieboldii* (syn. *Cinnamomum okinawense*) were collected from Tanabe city (Wakayama, Japan) in September 2021, and were separated into leaves, branches and stem barks. After shade-dried for several weeks, the materials were submitted to hydro-distillation for 3 hr with distilled water using a Clevenger-type apparatus. The obtained essential oils were stored at 4˚C until further analysis.

## Gas chromatography-mass spectrometry (GC/MS) analysis

Chemical characterization was performed by gas chromatograph coupled with mass spectrometer model QP2010 (Shimadzu, Kyoto, Japan). Essential oils were dissolved in acetone (2 μL/mL). This solution (1 μL) was injected in split mode (1:50 ratio) onto a DB-5MS column (30 m × 0.25 mm i.d. × 0.25 μm film thickness, Agilent, USA). The injection temperature was set at 270˚C. The oven temperature was started at 60˚C for 1 min after injection and then increased at 10˚C/min to 180˚C for 1 min, increased at 20˚C/min to 280˚C for 3 min followed by an increase at 20˚C/min to 325˚C, where the column was held for 20 min. Mass spectra were obtained in the range of 20 to 550 m/z. Essential oil components were identified based on a search (National Institute of Standards and Technology, NIST 14), the calculation of retention indices relative to homologous series of *n*-alkane, and a comparison of their mass spectra libraries with data from the mass spectra in the literature [16, 17].

## Antimicrobial assay

Broth microdilution assay was performed according to standard method of Japan Society of Chemotherapy [18] with slight modification. A stock solution of each essential oil (dissolved to a concentration of 40 mg/mL in DMSO) was diluted to 4 mg/mL by Mueller-Hinton II broth medium, followed by serial dilution by the medium to lower concentrations (2, 1, 0.5, 0.25, 0.125, 0.0625, 0.0313, 0.0156 and 0.0078 mg/mL). Thymol, a known antimicrobial agent, was dissolved and diluted in the same way to ensure microbial susceptibility (positive control). The oils were all tested in triplicate. *Staphylococcus aureus* NBRC 12732 was inoculated onto normal agar plates, and cultured for 24 hr at 35±1˚C. The bacterial suspensions were diluted by saline to obtain 0.5 McFarland turbidity equivalent (*ca*. $10^8$ colony forming units per mL (CFU/mL)), and were further diluted 10 times (*ca*. $10^7$ CFU/mL). 0.1 mL of essential oil-containing medium and 5 μL inoculum were added to sterile micro-titre plates. 10% (v/v) DMSO in the medium was used to determine if the solvent exhibited any antimicrobial effect (negative

control). The micro-titre plates were incubated for 18 to 24 hr at $35\pm1^\circ$C. Based on the opacity and color change in each well, minimum concentration capable of inhibiting the growth was determined.

## Results

### Machine learning of antimicrobial activity of chemical compounds

The classification models for antimicrobial compounds were successfully constructed in the semantic space. All machine learning algorithms showed good accuracies ranged from 84.3 to 85.4% in the five-fold cross-validation (S2 Table). In the following sections, SVM was adopted for further evaluations because it showed the best average accuracy.

The constructed model classified 726 MeSH compounds as active even though they were assumed to be inactive in the learning process. The top 10 MeSH compounds ranked by antimicrobial probability were shown in Table 1. Among the compounds, perillyl alcohol [19], daphnoretin [20], xanthohumol [21], rhodomyrtone [22], galbanic acid [23] and alpha-hederin [24] were previously reported to show antimicrobial activities. These compounds are potentially active, although they are not annotated as active compounds in the databases.

### Virtual screening of antimicrobial plants

Out of 2534 plant genera, 561 were predicted as active by the classifier (**S3 Table**). Among them, 164 were overlapped with antimicrobial plants listed in the review [10]. On the other hand, 265 genera in the review were predicted as inactive (sensitivity = 38.2%). The results were also shown as enrichment curve (Fig 2). The closer the curve is to the ideal curve, the higher the predictive performance of the model is. In the top 1% ranked plant genera (25 genera), WEBVS model correctly predicted 9 active genera, while 4.2 (1% of 429) active genera were expected to be included at random sampling (Table 2).

### Plant selection and extraction of essential oil

*Lindera* is a genus predicted as active (antimicrobial probability = 0.910), although it is not listed in the systematic review [10]. In fact, various pharmacological and biological properties of *Lindera* plants have been focused in many studies [25]. In this study, *Lindera triloba*, an endemic species in Japan, was selected for antimicrobial bioassay. The essential oils from branch and leaf of *Lindera triloba* were obtained by hydrodistillation with yields (v/w % on dry weight basis) of 0.36% and 0.46%, respectively (S4 Table).

**Table 1. The top 10 ranked compounds with higher antimicrobial probability.**

| Compound | Probability | Biological annotation |
|---|---|---|
| perillyl alcohol | 0.975 | antineoplastic agents, enzyme inhibitors |
| hydrazones | 0.944 | – |
| daphnoretin | 0.938 | antiviral agent, antineoplastic agent |
| xanthohumol | 0.938 | apoptosis inducer, antineoplastic agent, antiviral agent, diacylglycerol O-acyltransferase inhibitor, anti-HIV-1 agent |
| rhodomyrtone | 0.918 | – |
| calomel | 0.917 | – |
| dehydroabietinol | 0.915 | – |
| galbanic acid | 0.901 | – |
| naphthoquinones | 0.895 | – |
| alpha-hederin | 0.890 | anti-inflammatory agent |

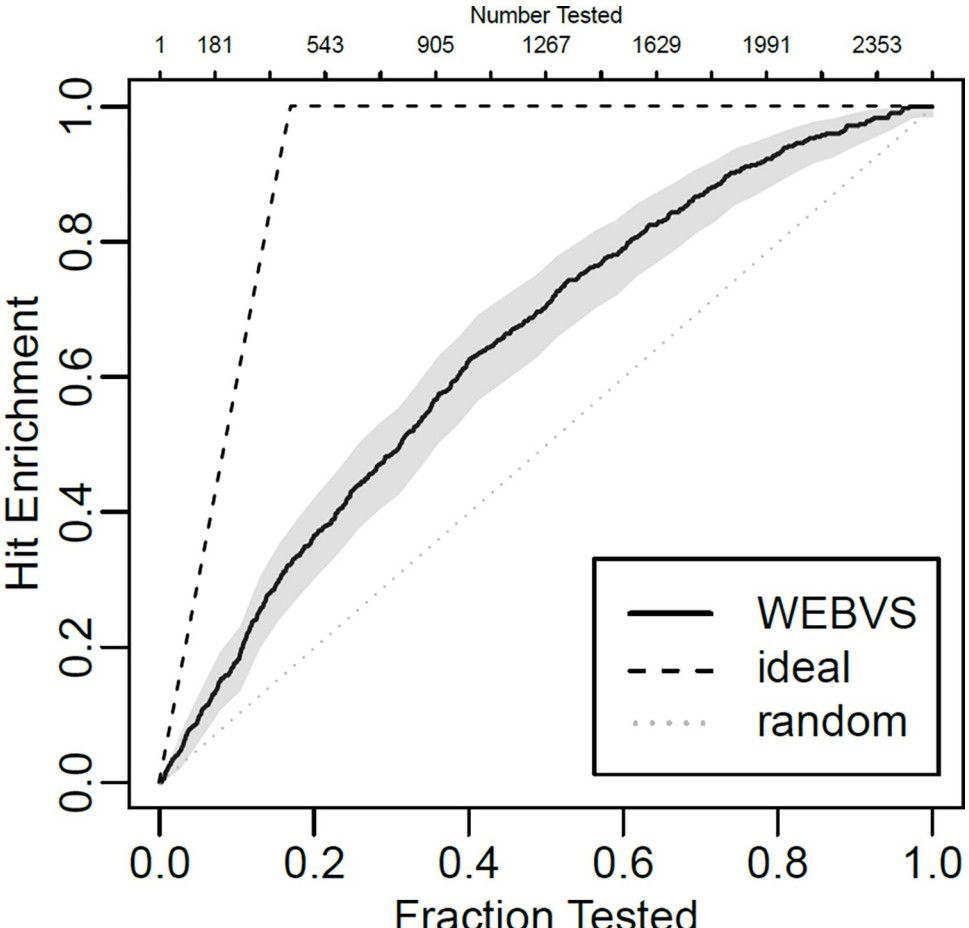

**Fig 2. Enrichment curve obtained by WEBVS.** The simultaneous 95 percent plus-adjusted sup-t confidence bands are colored in gray.

*Cinnamomum* is one of the genera with the most species investigated for antibacterial activity [10], and was also predicted as active (antimicrobial probability = 0.614) in this study. *Cinnamomum sieboldii*, a species grown wild in Japan, was also selected for antimicrobial assay. The essential oils from branch, leaf and stem bark of *Cinnamomum sieboldii* were obtained by hydrodistillation with yields of 0.80%, 0.64% and 0.58%, respectively (S4 Table).

## Chemical composition of selected essential oils

The chemical profile of investigated essential oils determined via GC/MS analysis, was presented in Table 3 and S4 Table. The main constituents of *Lindera triloba* branch oil were α-cadinol (9.4%), epi-α-muurolol (9.3%), camphor (9.1%), whereas those of the leaf oil were δ-cadinene (14.7%), α-cadinol (11.3%) and epi-α-muurolol (10.8%).

The main constituents of *Cinnamomum sieboldii* leaf oil were linalool (24.8%), cinnamaldehyde (19.1%), geranial (12.1%), whereas those from the other parts were linalool (branch: 51.2%, stem bark: 41.4%) followed by cinnamaldehyde (21.0%, 19.0%).

## Antimicrobial assay

The minimum inhibitory concentration (MIC) values against *S. aureus* were 1 mg/mL for *Lindera triloba* branch oil and 4 mg/mL for the leaf oil. The MIC value of *Cinnamomum sieboldii*

**Table 2.  The top 1% ranked plants with higher antimicrobial probability.**

| Genus | Probability |
|---|---|
| *Casearia* | 0.965 |
| *Lithospermum* | 0.956 |
| *Syngonanthus* | 0.956 |
| *Forsythia* | 0.938 |
| *Daphne* | 0.930 |
| *Biancaea* | 0.924 |
| *Ruta* | 0.918 |
| *Chelidonium* | 0.916 |
| *Sophora* | 0.914 |
| *Peganum* | 0.913 |
| *Spatholobus* | 0.911 |
| *Lindera* | 0.910 |
| *Ecballium* | 0.907 |
| *Carapa* | 0.901 |
| *Humulus* | 0.899 |
| *Garcinia* | 0.898 |
| *Alisma* | 0.897 |
| *Copaifera* | 0.895 |
| *Zanthoxylum* | 0.893 |
| *Boesenbergia* | 0.891 |
| *Kaempferia* | 0.890 |
| *Gardenia* | 0.890 |
| *Buddleja* | 0.888 |
| *Croton* | 0.887 |
| *Pentanema* | 0.886 |
| *Polygonum* | 0.886 |

Gray background indicates antimicrobial plants reviewed by Chassagne *et al*. [10]

oils from leaf, branch and stem bark were all 1 mg/mL (Table 4). These values are considered to be active with reference to Gibbons' paper which defined the essential oils having significant activity if the MIC is equal to or less than 5 μL/mL [26]. MIC for thymol (positive control) was 0.25 mg/mL, which was equivalent to literature data (0.03 v/v % [27]). No inhibition of bacterial growth was observed in the negative control.

**Table 3.  Major components of essential oils from *Lindera triloba* and *Cinnamomum sieboldii*.**

| Species | Parts | Major compounds identified (%)* |
|---|---|---|
| *Lindera triloba* | leaf | δ-cadinene (14.7), α-cadinol (11.3), epi-α-muurolol (10.8), α-muurolene (6.1), alloaromadendrene (6.0), β-bisabolene (6.0) |
| | branch | α-cadinol (9.4), epi-α-muurolol (9.3), camphor (9.1), limonene (8.3), bornyl acetate (7.5), δ-cadinene (7.1) |
| *Cinnamomum sieboldii* | leaf | linalool (24.8), cinnamaldehyde (19.1), geranial (12.1) |
| | branch | linalool (51.2), cinnamaldehyde (21.0), 1,8-cineole (11.8) |
| | stem bark | linalool (41.4), cinnamaldehyde (19.0), 1,8-cineole (10.3) |

Values in parentheses are the percentage of the total peak area obtained from the total ion current chromatogram.

**Table 4. Antimicrobial activity of essential oils from selected plants against *Staphylococcus aureus*.**

|  | *Lindera triloba* | | *Cinnamomum sieboldii* | | |
|---|---|---|---|---|---|
|  | leaf | branch | leaf | branch | stem bark |
| MIC (mg/mL) [a] | 4 | 1 | 1 | 1 | 1 |

[a] MIC: Minimum inhibitory concentration

## Discussion

Drug discovery and development is a long and costly process that takes years with an average cost of over $1–2 billion to be approved as a new drug [28]. Various technologies for miniaturization, lab automation and robotics have enabled pharma to perform bioassay targeting massive chemical compounds by means of high-throughput screening (HTS) [29]. However, application of HTS for identification of biologically active natural products remains a relatively uncommon activity because of requirement of expensive equipment and a variety of experimental obstacles such as sample unavailability (restricted season or location), degradation, precipitation and non-specific/off-target effects [30]. Therefore, computational approach is of great help in understanding the bioactivity of plant extracts composed of complex mixtures of phytochemicals. In this research, WEBVS method successfully classified antimicrobial plant extracts by capturing local context similarity between bioactive compounds and plant extracts.

The most important advantage of WEBVS is unnecessity of manual data curation that is costly and time-consuming process. Although recent studies [31, 32] showed good performance of QCAR-based model at predicting antimicrobial activity of essential oils, they have limitations in collecting new training data. WEBVS consists of simple and automated processes with public literature data which is regularly updated, indicating that the classification model is easily constructed and updated. Furthermore, WEBVS is suitable for large-scale exploration because it is applicable to all plants that appeared in literature data.

WEBVS also fits the idea of drug repositioning [33] that identifies new therapeutic uses for already-available drugs including approved, shelved and withdrawn drugs. To our knowledge, this is the first report on antimicrobial activity of *Lindera triloba* and *Cinnamomum sieboldii*. *Lindera triloba* is a deciduous shrub distributed on the Pacific side of the islands (Honshu, Shikoku and Kyushu) in Japan [34], and was reported to show insect anti-feeding activity [35]. In this research, GC/MS analysis of the essential oils revealed the presence of various sesquiterpene alcohols including α-cadinol and epi-α-muurolol (τ-cadinol). These alcohols were determined to be active by Su *et al.* [36], and are considered to contribute to the antimicrobial activity of *Lindera triloba*. *Cinnamomum sieboldii* is an evergreen arbor that used to be cultivated as a substitute for cassia (*Cinnamomum cassia*), and was used as traditional Japanese medicine in the 19th century. Watanabe and Goto reported that quantity of the essential oil compares favorably with that of cassia [37]. However, *Cinnamomum sieboldii* was removed from Japanese Pharmacopoeia (7th edition) in 1962 because the increasing import of low-cost cassia rendered it unnecessary as a substitute [38]. Both linalool and cinnamaldehyde, detected as main constituents of the essential oil in this study, were reported to show antimicrobial activity against *S. aureus* [27]. Further researches including clinical studies are needed to reconsider the medicinal use of *Cinnamomum sieboldii*.

Literature-based discovery, a text mining technique used to discover new knowledge implicitly present in scientific literature, has become widespread as scientific literature is growing at an exponential rate [39]. However, it has not been systematically explored in context with natural products [40]. Our WEBVS strategy can also be considered as an automated

literature-based discovery trying to build a knowledge bridge from chemistry area to the natural product area. Development of different literature-based models such as co-occurrence models and semantic models may also support the drug discovery and drug repositioning for natural products as well.

Finally, WEBVS has potential limitations. The first is that WEBVS cannot predict for a plant which has never been reported before. Approximately 13500 plant genera have been identified worldwide [41], but just 19% of them (2534 genera) were targeted in this study because of the lack of literature data. Combining WEBVS with phylogenetic analysis may be a promising approach because secondary metabolites of the plants are often similar within members of a clade [42]. The second limitation concerns the quantitativity. Any values in the text data did not influence the embedding, indicating that WEBVS is not suitable for quantitative prediction. However, it is generally difficult to combine quantitative activity data from multiple studies because the method and experimental conditions differ among them. Development of a relation extraction technique could help for integration and prediction of the quantitative activity data from full-text, tables and figures of the articles. The third limitation concerns chemical and bioactive variation due to environmental conditions. Various factors including temperature, carbon dioxide, lighting, ozone, soil water, soil salinity and soil fertility are known to affect plants' physiological and biochemical responses [43]. These factors may cause prediction error of WEBVS.

In conclusion, WEBVS is an efficient approach for exploring antimicrobial plant extracts. Application of WEBVS for other biological activities will be evaluated in future research.

## Supporting information

**S1 Table. Active compounds used for the machine learning of antimicrobial activity.**
(XLS)

**S2 Table. Accuracy result of various machine learning algorithms.**
(XLS)

**S3 Table. Antimicrobial probability of plant genera.**
(XLS)

**S4 Table. Chemical composition of essential oils from *Lindera triloba* and *Cinnamomum sieboldii*.**
(XLS)

## Acknowledgments

We sincerely thank Mr. & Mrs. Shimoyama (Monpetokuwa) and Mr. Nishida (Forestry cooperative of temple estate in Koya-san) for providing the plant samples used in this study. We appreciate the assistance of Kazuaki Sakaguchi, Sayo Sugimoto and Yuki Kishimoto for selection of the plant samples.

## Author Contributions

**Conceptualization:** Hiroaki Yabuuchi.

**Data curation:** Hiroaki Yabuuchi.

**Formal analysis:** Hiroaki Yabuuchi.

**Funding acquisition:** Hiroaki Yabuuchi, Kazuyuki Miyai.

**Investigation:** Hiroaki Yabuuchi, Kazuhito Hayashi.

**Methodology:** Akihiko Shigemoto, Yuhei Nomura.

**Project administration:** Shin-ichi Tokumoto, Kazuyuki Miyai.

**Resources:** Makiko Fujiwara, Yuhei Nomura.

**Software:** Yuhei Nomura, Mayumi Nakashima.

**Supervision:** Megumi Mori, Kazuyuki Miyai.

**Validation:** Kazuhito Hayashi, Takeshi Ogusu.

**Visualization:** Yuhei Nomura, Mayumi Nakashima.

**Writing – original draft:** Hiroaki Yabuuchi.

**Writing – review & editing:** Akihiko Shigemoto, Megumi Mori.

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
