## [Decision Letter · Decision Letter 0]

19 Apr 2023

PONE-D-23-09446Virtual screening of antimicrobial plant extracts by machine-learning classification of chemical compounds in semantic spacePLOS ONE

Dear Dr. Yabuuchi,

Thank you for submitting your manuscript to PLOS ONE. After careful consideration, we feel that it has merit but does not fully meet PLOS ONE’s publication criteria as it currently stands. Therefore, we invite you to submit a revised version of the manuscript that addresses the points raised during the review process.

We look forward to receiving your revised manuscript.

Kind regards,

Guadalupe Virginia Nevárez-Moorillón, Ph.D.

Academic Editor

PLOS ONE

Journal Requirements:

3. Please remove your figures from within your manuscript file, leaving only the individual TIFF/EPS image files, uploaded separately. These will be automatically included in the reviewers’ PDF.

Additional Editor Comments:

Please, consider the minor revisions suggested by the reviewers.

Reviewers' comments:

Reviewer's Responses to Questions

**Comments to the Author**

1. Is the manuscript technically sound, and do the data support the conclusions?

Reviewer #1: Yes

Reviewer #2: Yes

2. Has the statistical analysis been performed appropriately and rigorously? 

Reviewer #1: Yes

Reviewer #2: Yes

3. Have the authors made all data underlying the findings in their manuscript fully available?

Reviewer #1: Yes

Reviewer #2: Yes

4. Is the manuscript presented in an intelligible fashion and written in standard English?

Reviewer #1: Yes

Reviewer #2: Yes

5. Review Comments to the Author

Reviewer #1: Comment for the manuscript:

“Virtual screening of antimicrobial plant extracts by machine-learning classification of chemical compounds in semantic space”

Author have applied semantic space and word embedding algorithm to develop machine learning for classification of antimicrobial compouds . This a new type methods to develop machine learning model for classification.

In my opinion -

It is a well-written manuscript.

Methods are written explaning every step.

The results and figures are well explained.

The discussion and conclusion explain very well the outcomes of this manuscript.

Reviewer #2: The manuscript entitled ‘Virtual screening of antimicrobial plant extracts by machine-learning

classification of chemical compounds in semantic space’ described the evaluation of a new computational screening strategy for classifying bioactive compounds and plants in semantic space generated by word embedding algorithm.

The manuscript is well written. The study is relevant and important especially for researchers in countries where traditional knowledge about the medicinal uses of available plant has become limited or unavailable. In such situations applications like this can help explore the flora for drug discovery. It is also important for areas with rich traditional knowledge but where indigens are unwilling to give out information on important medicinal plants for research.

Unfortunately, the application does not address the issues of chemical and bioactive variation in plants due to factors such as geographical location of plant material, season of harvesting etc. that may affect predictions by the application.

Lines 236-238: A stronger reason HTS are uncommon is the cost involved in setting one up (acquisition, installation and maintenance). Revise.

Linea 248-249: why should plants that were previous investigated be subjects to WEBVS in its large-scale exploration? Is it necessary?

Linea 264: What else is there to consider about the medicinal potential of C. sieboldii when the authors had indicated that the plant is in short supply therefore its removal from the Japanese Pharmacopoeia? Revise.

Overall, it is a good manuscript worthy for consideration for publication after minor corrections.

6. PLOS authors have the option to publish the peer review history of their article (what does this mean?). If published, this will include your full peer review and any attached files.

Reviewer #1: **Yes: **Subhash Chandra

Reviewer #2: **Yes: **Gustav Komlaga

---

## [Author Response · Author response to Decision Letter 0]

27 Apr 2023

Journal Requirements:

RESPONSE: Thank you for your advice. We have checked the style requirements again, and uploaded figure files corrected by PACE.

RESPONSE: We have uploaded the code to GitHub, and added a description “R scripts and preprocessed literature data are available at https://github.com/yabuuchi-hiroaki/webvs” to “Data Availability” field.

3. Please remove your figures from within your manuscript file, leaving only the individual TIFF/EPS image files, uploaded separately. These will be automatically included in the reviewers’ PDF.

RESPONSE: We have removed the figures. We are sorry for forgetting to remove them.

RESPONSE: Thank you for your advice. We have checked the reference list again.

Reviewers' comments:

5. Review Comments to the Author

Reviewer #1: Comment for the manuscript:

“Virtual screening of antimicrobial plant extracts by machine-learning classification of chemical compounds in semantic space” Author have applied semantic space and word embedding algorithm to develop machine learning for classification of antimicrobial compouds . This a new type methods to develop machine learning model for classification.

In my opinion - It is a well-written manuscript. Methods are written explaning every step. The results and figures are well explained. The discussion and conclusion explain very well the outcomes of this manuscript.

RESPONSE: Thank you for dedicating your time to review our manuscript and for the positive opinion.

Reviewer #2: The manuscript entitled ‘Virtual screening of antimicrobial plant extracts by machine-learning classification of chemical compounds in semantic space’ described the evaluation of a new computational screening strategy for classifying bioactive compounds and plants in semantic space generated by word embedding algorithm.

The manuscript is well written. The study is relevant and important especially for researchers in countries where traditional knowledge about the medicinal uses of available plant has become limited or unavailable. In such situations applications like this can help explore the flora for drug discovery. It is also important for areas with rich traditional knowledge but where indigens are unwilling to give out information on important medicinal plants for research.

Unfortunately, the application does not address the issues of chemical and bioactive variation in plants due to factors such as geographical location of plant material, season of harvesting etc. that may affect predictions by the application.

RESPONSE: Thank you for dedicating your time to review our manuscript and for constructive comments. You have raised an important point in the last paragraph. We have incorporated your comments by p.18, lines 284-287.

Lines 236-238: A stronger reason HTS are uncommon is the cost involved in setting one up (acquisition, installation and maintenance). Revise.

RESPONSE: Thank you for your advice. We agree with you and have incorporated this suggestion (p.15, line 238).

Linea 248-249: why should plants that were previous investigated be subjects to WEBVS in its large-scale exploration? Is it necessary?

RESPONSE: We wrote "appeared in previous studies" to mean somewhat described in the literature data. However, as you pointed out, this phrase give an impression that the activity of plant is already investigated in previous studies. We have amended the manuscript to avoid reader's confusion (p.16, line 250).

Linea 264: What else is there to consider about the medicinal potential of C. sieboldii when the authors had indicated that the plant is in short supply therefore its removal from the Japanese Pharmacopoeia? Revise.

Overall, it is a good manuscript worthy for consideration for publication after minor corrections.

RESPONSE: Thank you for the suggestive question. We have rewritten the history and background of C. sieboldii with some references (from p.16 line 259 to p.17 line 263), and specified the need of further research including clinical studies (p.17 lines 265-266).

---

## [Editor Report · Decision Letter 1]

2 May 2023

Virtual screening of antimicrobial plant extracts by machine-learning classification of chemical compounds in semantic space

PONE-D-23-09446R1

Dear Dr. Yabuuchi,

We’re pleased to inform you that your manuscript has been judged scientifically suitable for publication and will be formally accepted for publication once it meets all outstanding technical requirements.

Kind regards,

Guadalupe Virginia Nevárez-Moorillón, Ph.D.

Academic Editor

PLOS ONE

Additional Editor Comments (optional):

I reviewed the corrections in the final document, as described in the cover letter by the corresponding author. Thank you for the edits and the manuscript can be accepted without further changes.
---

## [Editor Report · Acceptance letter]

3 May 2023

PONE-D-23-09446R1 

Virtual screening of antimicrobial plant extracts by machine-learning classification of chemical compounds in semantic space 

Dear Dr. Yabuuchi:

I'm pleased to inform you that your manuscript has been deemed suitable for publication in PLOS ONE. Congratulations! Your manuscript is now with our production department. 

Kind regards, 

on behalf of

Dr. Guadalupe Virginia Nevárez-Moorillón 

Academic Editor

PLOS ONE